# Tracking Multiple Targets Using Bearing-Only Measurements in Underwater Noisy Environments

**DOI:** 10.3390/s22155512

**Published:** 2022-07-24

**Authors:** Jonghoek Kim

**Affiliations:** Electronic and Electrical Department, Sungkyunkwan University, Suwon 03063, Korea; jonghoek@gmail.com

**Keywords:** underwater multiple target tracking, Gaussian mixture probability hypothesis density, GM-PHD, bearing-only tracking

## Abstract

This article handles tracking multiple targets using bearing-only measurements in underwater noisy environments. For tracking multiple targets in underwater noisy environments, the Gaussian Mixture Probability Hypothesis Density (GM-PHD) filter provides good performance with its low computational load. Bearing-only measurements are passive and do not provide position information of a target. Note that the nonlinearity of the bearing-only measurements can be handled by Extended Kalman Filters (EKF) when applying the GM-PHD filter. However, range uncertainty of the target is large for bearing-only measurements. Thus, a single EKF leads to poor performance when it is applied in the GM-PHD. In this article, every bearing measurement gives birth to multiple target samples, which are distributed considering the feasible range of the passive sensor. Thereafter, every target sample is updated utilizing the measurement update step of the EKF. In this way, we run multiple EKFs associated to multiple target samples, instead of running a single EKF. To the best of our knowledge, our article is novel in tracking multiple targets in noisy environments, using the observer with bearing-only measurements. The effectiveness of the proposed GM-PHD is verified utilizing MATLAB simulations.

## 1. Introduction

Multiple Target Tracking (MTT) is applied as multiple targets are tracked with a camera [1] or sonar sensors [2,3]. The goal of MTT is to conjecture the target information from a series of measurement sets.

In this article, we consider a single observer, which has a passive sonar (receiver) array for measuring the bearing of sound emitted from a target. Array signal processing algorithms, such as the MUltiple SIgnal Classification (MUSIC) algorithm, have been used to estimate the bearing of sound generated from a target [4,5,6,7,8].

For underwater MTT, passive sonars are desirable, since they work in a stealthy manner and consume little energy. However, background noise (interference) may generate bearing measurements that do not originate from a true target. In addition, the birth and death of the target result in uncertain data association between target and bearing measurement. Therefore, data association in noisy environments is required for tackling underwater multiple target tracking problems.

Bearing-only measurements can be utilized for tracking a moving target [9,10,11], such as a target with a constant velocity [12]. Processing bearing-only measurements is a nonlinear estimation problem; thus, it requires various nonlinear filtering techniques for its solution [13,14,15,16,17].

For processing bearing-only measurements, one can apply the Extended Kalman filter (EKF), which is the nonlinear version of the Kalman filter linearization about an estimate of the current mean and covariance [11,14,18,19,20]. In the EKF, the state transition and observation models do not have to be linear functions of the state but may instead be differentiable functions. The EKF has both low computational complexity and reasonable estimation accuracy; therefore, it has been widely used for tracking based on bearing-only measurements [11,14,19,20]. In our paper, we also use the EKF for processing bearing-only measurements.

For handling an observer tracking a target using bearing-only measurements, the Range-Parameterized Extended Kalman Filter (RPEKF) was developed [19,20]. The RPEKF is a Gaussian-sum filter with multiple EKFs, each initialized at an estimated target range. In this way, the RPEKF reduces the initial range estimation error. The RPEKF assumes that the maximum sensing range of the observer is known in advance. Inspired by the RPEKF [19,20], our paper also assumes that the maximum sensing range of an observer is known in advance.

The RPEKF is suitable for tracking a single target using bearing-only measurements, and the RPEKF does not consider noisy environments. Distinct from the RPEKF, our paper considers tracking multiple targets using bearing-only measurements in noisy environments.

For tracking multiple targets in noisy environments, the Multiple Hypotheses Tracking (MHT) [21] and Joint Probability Data Association (JPDA) [22,23] filters were developed. However, the computational burden of these methods increases as the number of targets or interference points increases.

Recently, the Random Finite Set (RFS) was developed for avoiding explicit associations between measurement and target [3,24,25]. The Probability Hypothesis Density (PHD) filter [3,24,25] was invented using the RFS. The PHD is computationally effective, since it avoids data association between target and measurement. The authors of [25] invented the Gaussian Mixture Probability Hypothesis Density (GM-PHD) filter, which is time-efficient for solving the data association problem. For tracking multiple targets that maneuver in noisy environments, ref. [26] developed a multiple model GM-PHD filter. As far as we know, no paper on MTT has considered tracking multiple targets using bearing-only measurements.

In our article, the observer measures the bearing of sound emitted from a target in real time. Note that the nonlinearity of the bearing-only measurements can be handled by the EKF [18] when applying the GM-PHD filter. However, range uncertainty of the target is large for bearing-only measurements. This results in poor performance when a single EKF is utilized in the GM-PHD.

Therefore, in this article, every bearing measurement gives birth to multiple target samples, which are distributed considering the feasible range of the passive sensor. Note that the feasible range of the passive sensor is available since we assume that the maximum sensing range of an observer is known in advance. Thereafter, every target sample is updated using the measurement update step of the EKF. In this way, we run multiple EKFs associated to multiple target samples instead of running a single EKF.

As far as we know, this paper is novel in tracking multiple targets in noisy environments, as the observer uses bearing-only measurements. The effectiveness of the proposed GM-PHD is verified utilizing MATLAB simulations.

The remainder of this article is organized as follows. Section 2 introduces the bearing-only target tracking as the preliminary information of this article. Section 3 addresses the GM-PHD filter for bearing-only target tracking. Section 4 discusses MATLAB simulations for demonstrating the performance of the proposed filter. Conclusions are addressed in Section 5.

## 2. Preliminary Information

We introduce the matrix notations utilized in this paper. Let C[i] define the *i*-th element in a column vector C. Let $I_{n}$ define the identity matrix with *n* rows and *n* columns. Let $0_{n}$ define the zero matrix with *n* rows and *n* columns. diag(a,b,c,…) defines a diagonal matrix having a,b,c,… as its diagonal elements in this order.

### Bearing-Only Target Tracking

Let $x_{k}$ define the target state at sampling-stamp *k*. The target’s location and velocity at sampling-stamp *k* are defined as
(1)$$\begin{array}{c} x_{k}={\left[x_{k}^{t},y_{k}^{t},{\dot{x}}_{k}^{t},{\dot{y}}_{k}^{t}\right]}^{\prime }. \end{array}$$

Suppose that the maximum speed of a target is known a priori. Let $v_{max}$ define the target’s maximum speed.

Let *T* define the sampling interval. Let $\left[a_{x},a_{y}\right]$ indicate the target’s acceleration, which is not known in advance. Considering a target with Constant Velocity (CV) motion, its process model is
(2)$$\begin{array}{c} x_{k+1}=F*x_{k}+\mu_{k}, \end{array}$$
where the state transition matrix F is
(3)$$\begin{array}{c} F=\left[\begin{array}{cc} 1 & T\\ 0 & 1 \end{array}\right]⊗I_{2}. \end{array}$$

In (Equation 2), $\mu_{k}$ defines the process noise with mean 0 and variance Q, where
(4)$$\begin{array}{c} Q=\Gamma *diag\left(\sigma_{a_{x}}^{2},\sigma_{a_{y}}^{2}\right)*\Gamma^{\prime }. \end{array}$$

Here, $diag(\sigma_{a_{x}}^{2},\sigma_{a_{y}}^{2})$ is a diagonal matrix having $\sigma_{a_{x}}^{2},\sigma_{a_{y}}^{2}$ as its diagonal elements in this order. Here, $\sigma_{a}$ presents the standard deviation for *a*. In addition, (Equation 4) utilizes
(5)$$\begin{array}{c} \Gamma =\left[\begin{array}{c} \frac{T^{2}}{2}*I_{2}\\ T*I_{2} \end{array}\right]. \end{array}$$

Let $\left[x_{k}^{o},y_{k}^{o}\right]$ define the 2D position of the observer at sampling-stamp *k*. In addition, $\left[{\dot{x}}_{k}^{o},{\dot{y}}_{k}^{o}\right]$ defines the 2D velocity of the observer at sampling-stamp *k*. In bearing-only target tracking, observer maneuvering is required for target localization [27,28]. This implies that the observer changes its velocity $\left[{\dot{x}}_{k}^{o},{\dot{y}}_{k}^{o}\right]$ in order to obtain the observability on the target position.

Let $r_{max}$ define the maximum sensing range of the observer’s sonar. In addition, the sonar’s minimum sensing range is $r_{min}$. Inspired by the RPEKF [19,20], we assume that both $r_{min}$ and $r_{max}$ are known in advance.

Once the relative distance between the target and the observer is within the interval $\left[0,r_{max}\right]$, the bearing of the target can be measured by the observer. The bearing measurement at sampling-stamp *k* is
(6)$$\begin{array}{c} b\left(x_{k}\right)=atan2\left(y_{k}^{t}-y_{k}^{o},x_{k}^{t}-x_{k}^{o}\right)+e_{k}. \end{array}$$

Here, atan2(y,x) is the phase (angle) of a complex number x+jy. In addition, $e_{k}$ defines the measurement noise having a Gaussian distribution with mean 0 and standard deviation R.

Ignoring the noise term in (Equation 6), we obtain
(7)$$\begin{array}{ccc} x_{k}^{t} & = & ∥\left(x_{k}^{t}-x_{k}^{o},y_{k}^{t}-y_{k}^{o}\right)∥*\cos\left(b\left(x_{k}\right)\right)+x_{k}^{o},\\ y_{k}^{t} & = & ∥\left(x_{k}^{t}-x_{k}^{o},y_{k}^{t}-y_{k}^{o}\right)∥*\sin\left(b\left(x_{k}\right)\right)+y_{k}^{o}. \end{array}$$


## 3. GM-PHD Filter

We utilize the GM-PHD filter in [25]. The multitarget state space is defined as F(X). At sampling-stamp *k*, target sample set $X_{k}$ and bearing measurement set $Z_{k}$ are defined as
(8)$$\begin{array}{c} X_{k}=\left\{x_{k,1},x_{k,2},\ldots ,x_{k,N(k)}\right\}\in F\left(X\right). \end{array}$$
(9)$$\begin{array}{c} Z_{k}=\left\{z_{k,1},z_{k,2},\ldots ,z_{k,M(k)}\right\}. \end{array}$$

Here, N(k) defines the number of target samples and M(k) defines the number of bearing measurements at sampling-stamp *k*.

Every $x_{k}\in X_{k}$ defines the target pose (position and velocity) addressed in (Equation 9). For a given state $X_{k-1}$ at sampling-stamp k−1, every $x_{k-1}\in X_{k-1}$ survives at sampling-stamp *k* with probability $P_{S}\left(x_{k-1}\right)$. Therefore, the survived target is represented using a RFS $S_{k|k-1}\left(x_{k-1}\right)$.

At sampling-stamp *k*, a new target appears by spontaneous birth or by spawning from an existing target at sampling-stamp k−1. In addition, *spontaneous birth sets* $\Gamma_{k}$ and *spawned target sets*$B_{k|k-1}\left(x_{k-1}\right)$ are modeled as RFS at sampling-stamp *k*. Birth and spawn of a new target from bearing measurements are introduced in Section 3.

At sampling-stamp *k*, $X_{k}$ is
(10)$$\begin{array}{c} X_{k}=\underset{\zeta \in X_{k-1}}{⋃}S_{k|k-1}\left(\zeta \right)⋃\left(\underset{\zeta \in X_{k-1}}{⋃}B_{k|k-1}\left(\zeta \right)\right)⋃\Gamma_{k}. \end{array}$$

In addition, at sampling-stamp *k*, a target is detected by bearing sensors with probability $P_{D}\left(x_{k}\right)$. Every target $x_{k}\in X_{k}$ generates its associated bearing measurement $b(x_{k})$ at sampling-stamp *k*. See (Equation 6) for the equation of $b(x_{k})$. In addition, the bearing sensor measures an interference point at sampling-stamp *k*. An interference point is modeled as a RFS $K_{k}$. Therefore, the RFS measurement set $Z_{k}$ is
(11)$$\begin{array}{c} Z_{k}=K_{k}⋃\left(\underset{x\in X_{k}}{⋃}b\left(x\right)\right). \end{array}$$

Let $p_{k}\left(X_{k}|Z_{1:k}\right)$ define the posterior density, $f_{k|k-1}$ define the transition density, and $g_{k}\left(Z_{k}|X\right)$ define the likelihood of every target. The posterior is calculated as
(12)$$\begin{array}{c} p_{k|k-1}\left(X_{k}|Z_{1:k-1}\right)=\\ \int f_{k|k-1}\left(X_{k}|X\right)p_{k-1}\left(X|Z_{1:k-1}\right)\mu \left(dX\right). \end{array}$$
(13)$$\begin{array}{c} p_{k}\left(X_{k}|Z_{1:k}\right)=\frac{g_{k}\left(Z_{k}|X_{k}\right)p_{k|k-1}\left(X_{k}|Z_{1:k-1}\right)}{\int g_{k}\left(Z_{k}|X\right)p_{k|k-1}\left(X|Z_{1:k-1}\right)\mu \left(dX\right)}. \end{array}$$

Here, μ is an appropriate reference measure on F(X) [29].

It is assumed that an interference is independent of target-originated measurements. In addition, it is assumed that an interference point and predicted RFS are distributed using a Poisson distribution.

The survival and detection probabilities do not depend on the state vector; hence, we utilize $P_{S,k}$ and $P_{D,k}$ instead of $P_{S}\left(x_{k}\right)$ and $P_{D}\left(x_{k}\right)$, respectively.

Let N(x;m,P) denote the Gaussian density with mean *m* and covariance *P*. The birth intensity function of a newborn target at sampling-stamp *k* is
(14)$$\begin{array}{c} \gamma_{k}\left(x\right)=\sum_{i=1}^{J_{\gamma ,k}}w_{\gamma ,k}^{\left(i\right)}N\left(x;m_{\gamma ,k}^{\left(i\right)},P_{\gamma ,k}^{\left(i\right)}\right) \end{array}$$
where $J_{\gamma ,k}$ represents the number of newborn Gaussian components, and $w_{\gamma ,k}^{\left(i\right)}$, $m_{\gamma ,k}^{\left(i\right)}$, and $P_{\gamma ,k}^{\left(i\right)}$ are the weight, mean, and covariance, respectively, of the *i*-th newborn Gaussian component.

Let $v_{k|k}$ define the posterior density intensity at sampling-stamp *k*. The posterior intensity propagates utilizing
(15)$$v_{k|k-1}\left(x\right)=\int P_{S,k}f_{k|k-1}\left(x|\zeta \right)v_{k-1}\left(\zeta \right)d\zeta +\gamma_{k}\left(x\right).$$
(16)$$\begin{array}{cc} v_{k|k}\left(x\right)=\left(1-P_{D,k}\right)v_{k|k-1}\left(x\right) & +\\ \sum_{z\in Z_{k}}\frac{P_{D,k}g_{k}\left(z|x\right)v_{k|k-1}\left(x\right)}{\lambda_{k}c_{k}\left(z\right)+\int P_{D,k}g_{k}\left(z|\zeta \right)v_{k|k-1}\left(\zeta \right)d\zeta }. \end{array}$$

Here, $\lambda_{k}$ is the average number of interference points at sampling-stamp *k*. In addition, $c_{k}\left(z\right)$ defines the probability distribution of every interference point.

In the proposed GM-PHD, the prediction step is linear and the measurement update step is nonlinear. Thus, we only have to linearize the measurement update step. In our paper, the EKF is used for the measurement update step, since the EKF has both low computational complexity and reasonable estimation accuracy [11,14,19,20].

We perform the measurement update, once we obtain the newest measurement set $Z_{k}$ at sampling-stamp *k*. Since the measurement model in (Equation 6) is not linear, we can utilize the EKF in [18] for the measurement update step.

Under the GM-PHD [25,30], (Equation 17) is replaced by
(17)$$\begin{array}{c} v_{k|k-1}\left(x\right)=\sum_{i=1}^{J_{k|k-1}}w_{k|k-1}^{\left(i\right)}N\left(x;m_{k|k-1}^{\left(i\right)},P_{k|k-1}^{\left(i\right)}\right)=v_{S,k|k-1}\left(x\right)+\gamma_{k}\left(x\right) \end{array}$$
where
(18)$$\begin{array}{c} v_{S,k|k-1}\left(x\right)=\sum_{i=1}^{J_{k-1}}P_{S,k}w_{k-1}^{\left(i\right)}N\left(x;m_{k|k-1}^{\left(i\right)},P_{k|k-1}^{\left(i\right)}\right). \end{array}$$

Here, $J_{k-1}$ is the number of components of the intensity at sampling-stamp k−1. Moreover, we utilize
(19)$$\begin{array}{c} m_{k|k-1}^{\left(i\right)}=F*m_{k-1|k-1}^{\left(i\right)}. \end{array}$$

In addition, we utilize
(20)$$\begin{array}{c} P_{k|k-1}^{\left(i\right)}=F*P_{k-1|k-1}^{\left(i\right)}*F^{\prime }+Q. \end{array}$$

Further, (Equation 19) is replaced by
(21)$$\begin{array}{c} v_{k|k}\left(x\right)=\left(1-P_{D,k}\right)v_{k|k-1}\left(x\right)+\sum_{i=1}^{J_{k|k-1}}w_{k|k}^{\left(i\right)}\left(z\right)N\left(x;m_{k|k}^{\left(i\right)},P_{k|k}^{\left(i\right)}\right). \end{array}$$

In (Equation 24), we use
(22)$$\begin{array}{c} w_{k|k}^{\left(i\right)}\left(z\right)=\frac{P_{D,k}g_{k}\left(z|m_{k|k-1}^{\left(i\right)}\right)w_{k|k-1}^{\left(i\right)}}{\lambda_{k}c_{k}\left(z\right)+P_{D,k}\sum_{l=1}^{J_{k|k-1}}g_{k}\left(z|m_{k|k-1}^{\left(l\right)}\right)w_{k|k-1}^{\left(l\right)}}. \end{array}$$

Here, one uses
(23)$$\begin{array}{c} g_{k}\left(z|m_{k|k-1}^{\left(i\right)}\right)=N\left(z;b\left(m_{k|k-1}^{\left(i\right)}\right),S_{k}^{\left(i\right)}\right), \end{array}$$
where
(24)$$\begin{array}{c} S_{k}^{\left(i\right)}=H*P_{k|k-1}^{\left(i\right)}*H^{\prime }+R. \end{array}$$

In (Equation 24), we use
(25)$$\begin{array}{c} m_{k|k}^{\left(i\right)}=m_{k|k-1}^{\left(i\right)}+K\left(z_{k,g}-b\left(m_{k|k-1}^{\left(i\right)}\right)\right). \end{array}$$

Here, K is
(26)$$\begin{array}{c} K=P_{k|k-1}^{\left(i\right)}*H^{\prime }*{\left(S_{k}^{\left(i\right)}\right)}^{-1}. \end{array}$$

In addition, $H=\frac{\partial b(x)}{\partial x}$ where $x=m_{k|k-1}^{\left(i\right)}$.

In (Equation 24), we further use
(27)$$\begin{array}{c} P_{k|k}^{\left(i\right)}=\left(I-K*H\right)*P_{k|k-1}^{\left(i\right)}, \end{array}$$
where I is the identity matrix. The detailed derivation of (Equation 24) is addressed in [25,30].

To reduce the computational load, Gaussian components must be pruned and merged [30]. Then, the posterior intensity is represented as
(28)$$\begin{array}{c} v_{k|k}\left(x\right)=\sum_{i=1}^{J_{k}}w_{k}^{\left(i\right)}N\left(x;m_{k|k}^{\left(i\right)},P_{k|k}^{\left(i\right)}\right). \end{array}$$

Here, $w_{k}^{\left(i\right)}$ is the weight of Gaussian distribution and $J_{k}$ is the number of components of the intensity at sampling-stamp *k*.

### Birth or Spawn of New Target Samples Based on Bearing Measurements

Recall that at every sampling-stamp *k*, a new target sample is generated by spontaneous birth or by spawning from an existing target sample at sampling-stamp k−1. Spontaneous birth sets $\Gamma_{k}$ and spawned target sets $B_{k|k-1}\left(x_{k-1}\right)$ are modeled as RFS at sampling-stamp *k*.

In this article, the observer measures the bearing of sound emitted from a target in real time. Bearing-only measurements are passive and do not provide position information of a target. Therefore, in this article, every bearing measurement gives birth to multiple target samples, which are distributed considering the feasible range of the passive sensor.

We update the mean *m* and covariance *P* of the newly born targets in $\Gamma_{k}$ as follows. Suppose that we generate *B* target samples using $z_{k,g}$ (g∈{1,2,…,M(k)}), the *g*-th bearing measurement at sampling-stamp *k*. Let $t_{b}$ (b∈{1,2,…,B}) define the *b*-th target sample generated on the bearing line at sampling-stamp *k*.

The feasible range of the true target is in the interval $\left[r_{min},r_{max}\right]$. This interval is divided into *B* subintervals $\left[r_{min}+\left(b-1\right)*sR,r_{min}+b*sR\right]$, where $sR=\frac{r_{max}-r_{min}}{B}$ (b∈{1,2,…,B}).

The range for the center of every subinterval is $r_{b}=r_{min}+\left(b-0.5\right)*sR$ (b∈{1,2,…,B}). In addition, the velocity of every newly born target sample is zero. Under (Equation 7), the mean *m* of a newly born target sample $t_{b}$ (b∈{1,2,…,B}) is
(29)$$\begin{array}{c} m\left(t_{b}\right)=\left[\begin{array}{c} r_{b}*\cos\left(z_{k,g}\right)+x_{k}^{o}\\ r_{b}*\sin\left(z_{k,g}\right)+y_{k}^{o}\\ 0\\ 0 \end{array}\right] \end{array}$$
where $r_{b}=r_{min}+\left(b-0.5\right)*sR$.

We next derive the covariance *P* of $t_{b}$. Under (Equation 6), the variance of the bearing error is $R^{2}$. In addition, since every subinterval length is sR, the variance of the range error is $\frac{sR^{2}}{4}$.

The error covariance related to the 2D position of $t_{b}$ (b∈{1,2,…,B}) is
(30)$$\begin{array}{c} P_{s}\left(t_{b}\right)=D*diag\left(\frac{sR^{2}}{4},R^{2}\right)*D^{\prime } \end{array}$$
where D is
(31)$$\begin{array}{c} D=\left[\begin{array}{c} \frac{\partial m\left(t_{b}\right)\left[1\right]}{\partial r_{b}},\frac{\partial m\left(t_{b}\right)\left[1\right]}{\partial z_{k,g}}\\ \frac{\partial m\left(t_{b}\right)\left[2\right]}{\partial r_{b}},\frac{\partial m\left(t_{b}\right)\left[2\right]}{\partial z_{k,g}} \end{array}\right]. \end{array}$$

Under (Equation 33), we further obtain
(32)$$\begin{array}{c} D=\left[\begin{array}{c} \cos\left(z_{k,g}\right),-r_{b}*\sin\left(z_{k,g}\right)\\ \sin\left(z_{k,g}\right),r_{b}*\cos\left(z_{k,g}\right) \end{array}\right]. \end{array}$$

The covariance *P* of $t_{b}$ (b∈{1,2,…,B}) is derived as
(33)$$\begin{array}{c} P\left(t_{b}\right)=\left[\begin{array}{c} P_{s}\left(t_{b}\right),0_{2}\\ 0_{2},diag\left(v_{max}^{2},v_{max}^{2}\right) \end{array}\right]. \end{array}$$

In the case where the relative distance between a newly born target sample, e.g., $t_{b}$ (b∈{1,2,…,B}), and an existing target sample, e.g., $x_{k-1}$, is less than a certain threshold α>0, we say that the born target sample $t_{b}$ is *spawned* from the existing target sample $x_{k-1}$. This implies that we have $t_{b}\in B_{k|k-1}\left(x_{k-1}\right)$.

We update the mean *m* and covariance *P* of the newly spawned target sample $t_{b}\in B_{k|k-1}\left(x_{k-1}\right)$ as follows. We derive the covariance *P* of $t_{b}$ using (Equation 36). In addition, we use
(34)$$\begin{array}{c} m\left(t_{b}\right)=\left[\begin{array}{c} x_{k-1}\left[1\right]\\ x_{k-1}\left[2\right]\\ 0\\ 0 \end{array}\right]. \end{array}$$

## 4. MATLAB Simulations

We address MATLAB simulation results for verifying the effectiveness of the proposed GM-PHD filter. Target bearings are measured at every dt=1 s. In addition, the simulation running time is 500 s.

Target detection probability is 0.98. The target’s maximum speed is $v_{max}=30$ m/s. Considering the sonar sensing range, we set $r_{max}=$ 10,000 m. In addition, the sonar’s minimum sensing range is set to $r_{min}=500$ m. The standard deviation of bearing measurement noise is 1 degree.

In bearing-only target tracking, observer maneuvering is required for improving the tracking convergence [27,28]. Since the observer uses bearing-only measurements, the tracking convergence depends on the bearing rate of a target with respect to the observer. In other words, the geometry of a target with respect to the observer determines the tracking convergence of the target.

In the simulations, the observer maneuvers as follows. The observer moves with a constant speed $v_{obs}=10$ m/s, and it starts from the origin. Before 125 s, the observer moves with course 0 degree measured clockwise from the east direction. From 125 to 250 s, the observer moves with course 135 degrees measured clockwise from the east direction. From 250 to 375 s, the observer moves with course 0 degree measured clockwise from the east direction. After 375 s, the observer moves with course 90 degrees measured clockwise from the east direction.

See Figure 1 for the simulation scenario. In this figure, the observer’s path is marked with yellow colors. Every target moves with a constant velocity. In Figure 1, we present the direction of target motion using symbols. A circle symbol indicates the start point of a target motion and a diamond symbol indicates the end point of a target motion.

Since the sonar’s maximum sensing range is 10 km, we deploy a target initially so that it can be detected by the observer. The position of each target at every second is marked with distinct colors. Target 1 (red) starts from [7500, 7500] and moves with velocity [3, −5] in m/s. Target 2 (blue) starts from [−2500, −2500] and moves with velocity [0, 5] in m/s. Targets 1 (red) and 2 (blue) are generated at sampling-stamp 0, and target 3 (green) is spawned from target 1 at 30 s. Target 3 (green) moves with velocity [−5, −2] in m/s.

For clear presentation of the observer’s path, Figure 1 simulates the case where there is no interference point. In other words, Figure 1 shows the simulation result without interference points. In this figure, the estimated target position is marked with magenta triangles. Before the observer maneuver at 125 s, true target positions are not tracked by target estimations (magenta triangles). However, as time goes on, true target positions are tracked by target estimations.

Considering the simulation of the scenario in Figure 1, Figure 2 shows the simulation result with interference points. Recall that $\lambda_{k}$ (Equation 19) is the average number of interference points at sampling-stamp *k*. We set $\lambda_{k}=1$, which implies that, on average, one interference point is generated at each sampling-stamp. In Figure 2, the estimated target position is marked with magenta triangles. Due to interference points, many false targets are generated surrounding the observer. Before the observer maneuver at 125 s, true target positions are not tracked by target estimations (magenta triangles). However, as time goes on, true target positions are tracked by target estimations.

Considering the simulation of Figure 2, Figure 3 shows the bearing (in radians) change with respect to time (in seconds). In this figure, true target bearing measurements are marked with circles. In addition, interference points are marked with crosses. The estimated target bearing is marked with magenta circles. Before the observer maneuver at 125 s, true target bearings are not tracked by the estimated target bearing (magenta circles). However, as time goes on, true target bearings are tracked by estimated target bearings.

### 4.1. Monte-Carlo (MC) Simulations

A total of 100 Monte-Carlo (MC) simulations are utilized for verifying the outperformance of the proposed filter. Let $M_{C}=100$ denote the number of total MC simulations. We set $\lambda_{k}=1$, which implies that, on average, one interference point is generated at each sampling-stamp. We generate B=50 target samples using every bearing measurement at every sampling-stamp. It takes 14 s to run a single MC simulation.

Tracking accuracy is checked with the cardinality (number of targets) and the Optimal Subpattern Assignment (OSPA) metric [31,32]. We derive the cardinality as the mean of the estimated number of targets under $M_{C}=100$ MC simulations. We also derive the mean of OSPA using $M_{C}=100$ MC simulations.

Recall that $J_{k}$ in (Equation 31) is the number of components of the intensity. The cardinality at sampling-stamp *k* is defined as
(35)$$\begin{array}{c} card_{k}=\frac{\sum_{j=1}^{M_{C}}J_{k}}{M_{C}}. \end{array}$$

The true cardinality at sampling-stamp *k* indicates the true number of targets detected at sampling-stamp *k*.

We briefly address the Optimal Subpattern Assignment (OSPA) metric in [31,32]. Let $d^{\left(c\right)}\left(x,y\right)=min(c,∥x-y∥)$ denote the distance between x and y. Here, *c* denotes a cut-off parameter in [31,32]. In our simulations, we use $c=2*10^{4}$.

Let $M\left(k\right)=\{m_{1}\left(k\right),m_{2}\left(k\right),\ldots ,m_{J_{k}}\left(k\right)\}$ denote the set of target position estimates at sample-step *k*. Recall that $J_{k}$ denotes the number of components of the intensity. Suppose that there are N(k) targets at sample-step *k*. Let $\mathrm{Tgt}\left(k\right)=\{{\mathrm{tgt}}_{1}\left(k\right),{\mathrm{tgt}}_{2}\left(k\right),\ldots ,{\mathrm{tgt}}_{N(k)}\left(k\right)\}$ denote the true target positions at sample-step *k*. The OSPA distance between the target estimates M(k) and the true target positions Tgt(k) is defined as
(36)$$\begin{array}{c} ospa_{d}\left(k\right)=\frac{1}{N(k)}\left(min_{q\in P_{N(k)}}\sum_{i=1}^{J_{k}}d^{\left(c\right)}\left(m_{i}\left(k\right),{\mathrm{tgt}}_{q(i)}\left(k\right)\right)+c\left(N\left(k\right)-J_{k}\right)\right). \end{array}$$

Here, $P_{N(k)}$ denotes the set of permutations on {1,2,…,N(k)}. The OSPA metric can be computed efficiently by using the Hungarian method for optimal point assignment [31,32].

The mean of OSPA at sample-step *k* is
(37)$$\begin{array}{c} ospa_{k}=\frac{\sum_{j=1}^{M_{C}}ospa_{d}\left(k\right)}{M_{C}}. \end{array}$$

See Figure 4 for the OSPA $ospa_{k}$ and the cardinality $card_{k}$ of the proposed GM-PHD. Before 30 s pass, there are two targets (Targets 1 and 2). Target 3 is spawned from target 1 at 30 s. In bearing-only target tracking, observer maneuver is required for target localization [27,28]. The observer maneuvers at 125 s. See that the OSPA decreases gradually after the observer maneuver. In Figure 4, True plots the number of bearing measurements associated with true targets. In addition, Estimate is the mean of the estimated number of targets using 100 MC simulations. As time goes on, OSPA gradually approaches to zero and Estimate converges to True in cardinality. In other words, the cardinality error converges to almost zero as time goes on.

#### 4.1.1. Comparison with the Case Where a Single EKF Is Used in the GM-PHD

Note that the nonlinearity of the bearing-only measurements can be handled by the EKF when applying the GM-PHD filter. However, range uncertainty of the target is large for bearing-only measurements ($r_{max}=10$ km in MATLAB simulations). This results in poor performance when a single EKF is used in the GM-PHD. Setting B=1 corresponds to the case where we apply a single EKF in the GM-PHD. As we increase *B*, we distribute more target samples considering the maximum sensing range of the bearing sensor. Consider the scenario in Figure 2. We next check the effect of changing *B*, the number of target samples generated using every bearing measurement at every sampling-stamp.

Considering the case where B=1 is utilized, Figure 5 plots the OSPA and the cardinality of the proposed GM-PHD. Compared with the case where B=50 is used (Figure 4), the estimation accuracy decreases considerably. This shows the effectiveness of the proposed sample distribution approach used in our paper. It takes 6 s to run a single MC simulation, as we utilize B=1. Compared with the case where B=50 is used (Figure 4), the computational load decreases.

#### 4.1.2. The Effect of Changing *B*

Considering the case where B=10 is utilized, Figure 6 plots the OSPA $ospa_{k}$ and the cardinality $card_{k}$ of the proposed GM-PHD. Compared with the case where B=1 is used (Figure 5), the estimation accuracy increases considerably. It takes 6 s to run a single MC simulation. Compared with the case where B=50 is used, the computational load decreases.

Considering the case where B=20 is utilized, Figure 7 plots the OSPA and the cardinality of the proposed GM-PHD. Compared with the case where B=1 is used (Figure 5), the estimation accuracy increases considerably. It takes 7 s to run a single MC simulation. Compared with the case where B=50 is used, the computational load decreases.

Table 1 shows the computational load (running time for one MC simulation) as *B* varies. Recall that setting B=1 corresponds to the case where we apply a single EKF in the GM-PHD. Table 1 shows that as *B* increases, the computational load increases in general. However, considering both the computational load and the estimation accuracy (Figure 6), setting B=10 is desirable.

## 5. Conclusions

This article handles tracking multiple targets using bearing-only measurements in underwater noisy environments. We apply GM-PHD filters to solve our MTT problem based on bearing-only measurements. Since bearing-only measurements do not provide position information of a target, we make every bearing measurement give birth to *B* target samples, which are distributed considering the feasible range of the passive sensor. The effectiveness of the proposed GM-PHD is verified utilizing MATLAB simulations.

In the future, we will extend the proposed tracking filter so that one can handle the case where multiple observers are used for tracking multiple targets in noisy environments under bearing-only measurements. As we use multiple observers, false targets may appear as bearing lines of multiple observers’ intersects. Thus, we require filters for removing false targets effectively.

## Figures and Tables

**Figure 1 sensors-22-05512-f001:**
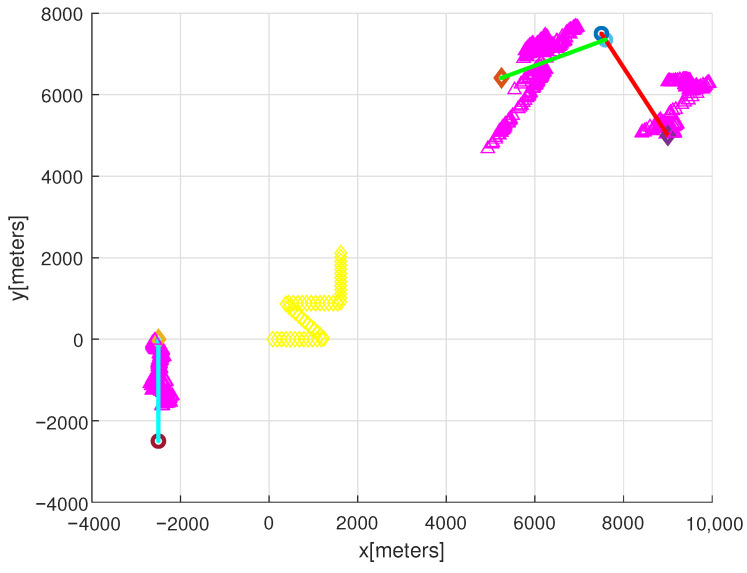
For clear presentation of the observer’s path, this figure simulates the case where there is no interference point. The observer’s path is marked with yellow colors. Target 1 (red) starts from [7500, 7500] and moves with velocity [3, −5] in m/s. Target 2 (blue) starts from [−2500, −2500] and moves with velocity [0, 5] in m/s. Targets 1 (red) and 2 (blue) are generated at sampling-stamp 0, and target 3 (green) is spawned from target 1 at 30 s. Target 3 (green) moves with velocity [−5, −2] in m/s. We present the direction of target motion using symbols. A circle symbol indicates the start point of a target motion, and a diamond symbol indicates the end point of a target motion. The estimated target position is marked with magenta triangles.

**Figure 2 sensors-22-05512-f002:**
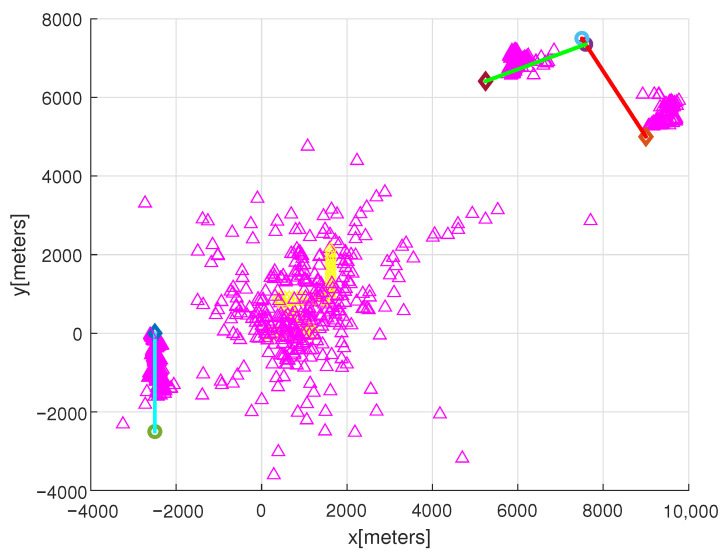
This figure shows the simulation result with interference points. We set $\lambda_{k}=1$, which implies that, on average, one interference point is generated at each sampling-stamp. The estimated target position is marked with magenta triangles. Before the observer maneuver at 125 s, true target positions are not tracked by target estimations (magenta triangles). However, as time goes on, true target positions are tracked by target estimations.

**Figure 3 sensors-22-05512-f003:**
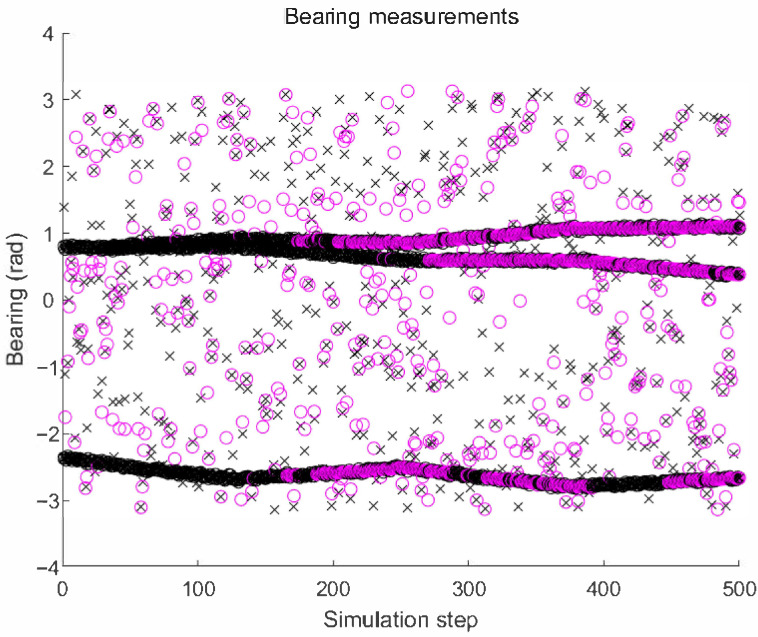
The bearing (in radians) change with respect to time (in seconds). In this figure, true target bearings are marked with circles. In addition, interference points are marked with crosses. The estimated target bearing is marked with magenta circles. Before the observer maneuver at 125 s, true target bearings are not tracked by the estimated target bearing (magenta circles). However, as time goes on, true target bearings are tracked by estimated target bearings.

**Figure 4 sensors-22-05512-f004:**
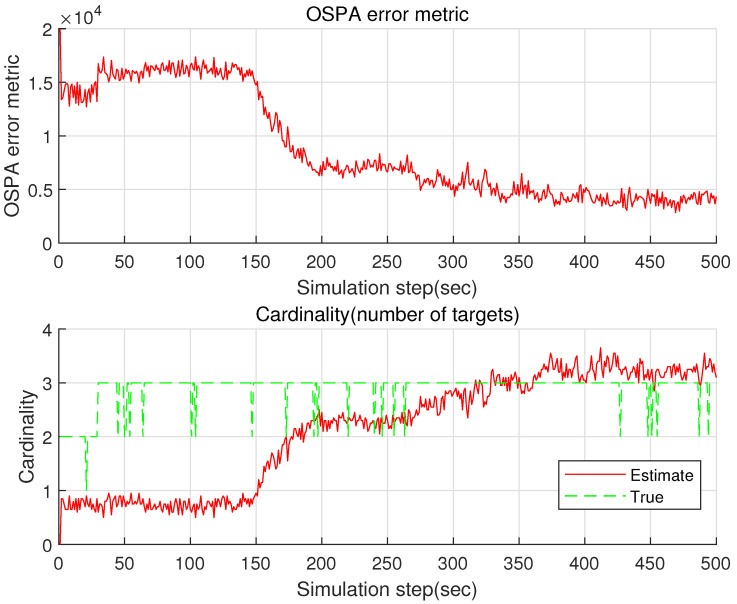
The OSPA and the cardinality of the proposed GM-PHD in the case where B=50 is utilized. As time goes on, OSPA gradually approaches to zero and Estimate converges to True in cardinality.

**Figure 5 sensors-22-05512-f005:**
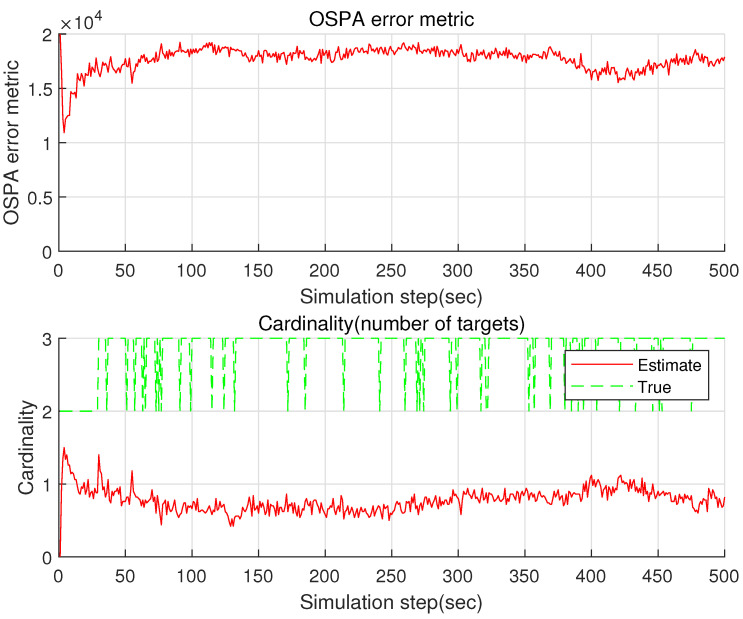
The OSPA and the cardinality of the proposed GM-PHD in the case where B=1 is utilized. Setting B=1 corresponds to the case where we apply a single EKF in the GM-PHD. Compared with the case where B=50 is used (Figure 4), the estimation accuracy decreases considerably.

**Figure 6 sensors-22-05512-f006:**
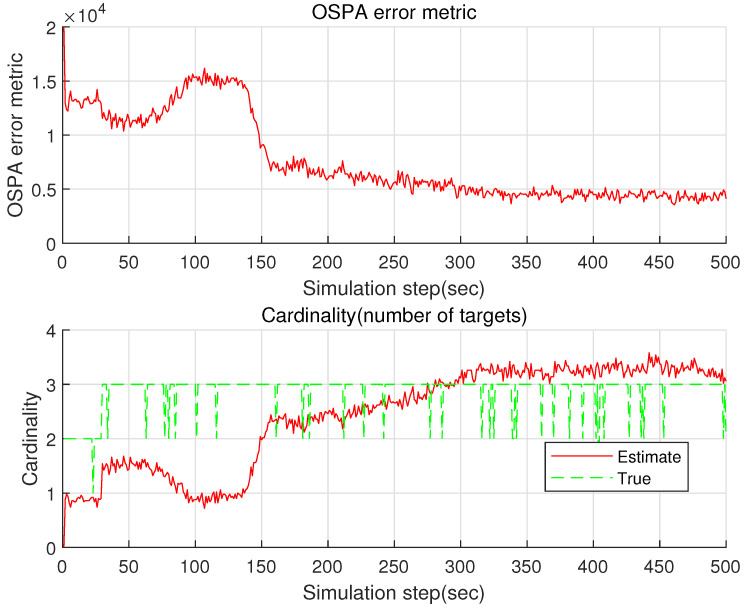
The OSPA and the cardinality of the proposed GM-PHD in the case where B=10 is utilized.

**Figure 7 sensors-22-05512-f007:**
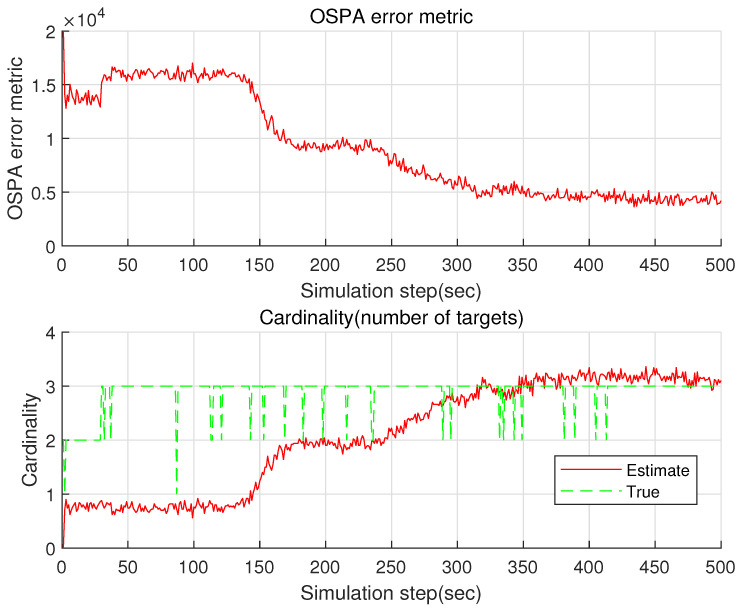
The OSPA and the cardinality of the proposed GM-PHD in the case where B=20 is utilized.

**Table 1 sensors-22-05512-t001:** Computational load analysis.

*B*	OneMCtime
1	6 s
10	6 s
20	7 s
50	14 s

## Data Availability

Not applicable.

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
