# Peer review of "Tracking Multiple Targets Using Bearing-Only Measurements in Underwater Noisy Environments"

_sensors, 2022, doi:10.3390/s22155512_

Round 1

Reviewer 1 Report

Comments to the Author:

This article describes a novel implementation of the GM-PHD filter for tracking multiple targets using bearing-only measurements in cluttered environments. Every bearing measurement creates multiple target samples distributed considering the range of the sensor. 

- The paper is well written and understandable. 

- This paper relies significantly on references throughout the methods section. It would be helpful to the reader to provide more background on teh GM-PHD filter.

- Readers familiar with target tracking are likely to understand the cardinality and OSPA metric. It would be good to provide definitions of these values instead of only a citation. 

- How do the computational requirements of this algorithm compare to the cited works?

- Is there a method for finding the optimal value of B (the division of the sensor range)?

- For the Matlab simulations, are the targets moving at constant velocity? Is their initial velocity randomly chosen between 0 and v_max=30 m/s?

- In the simulation results figures, labeling the direction of target motion would be helpful.

- In Figures 1 and 2, it appears that the red and blue targets are accurately tracked, but the green target is not. Do the authors have any comments on this?

Author Response

Thank you very much for your comments. The response to Reviewer 1 is attached.

Reviewer 2 Report

The paper arises an actual topic of sound source localization and tracking in underwater acoustics. Unfortunately, the paper cannot be published in the present form.

1. Figure captions do not help us to understand what is actually plotted. They are describing some conclusion from the figure, but first some initial legend should be given about each figure.

2. There is an extended math section in the paper, but the fact that the receiver is moving by itself is nearly missing in it. This essential information appears in the simulations section.

3. Please explicitly explain the ‘clutter’ term and the physics lying behind that. Clutter is usually studied in the context of active sonar, and in means a set of effects like random scattering and multipath propagation. Passive sonars suffer from background noise most of all. This should be called just noise or interference.

Thus, major revision is what is actually needed for this paper.

Author Response

Thank you very much for your comments. The response to Reviewer 2 is attached.

Round 2

Reviewer 2 Report

The paper has become much clearer after the revision.

One more issue should be discussed. The proposed algorithm starts working from the point, when another external algorithm has done the signal processing job, resulting in primary detection. In my opinion, the reader needs an insight which kind of algorithm can act on the first stage of processing. Such algorithms can be quite simple like filtering and threshold comparison, and really sophisticated. Take for example synthetic aperture sonar, which is aware of the observer movement. I have found that the presented research is somehow correlated with the:

Rodionov, A.A., Semenov, V.Y., Savel’ev, N.V. et al. Localization of a Moving Acoustic Source Using Incoherent Aperture Synthesis with Simultaneous Suppression of Interference. Radiophys Quantum El 63, 501–510 (2020)

DOI 10.1007/s11141-021-10074-y

(not my paper)

Author Response

(The authors gave the same response as above.)
